## [Peer Review File · EMBO Reports]

Human CRAMP1 specifically promotes the expression of histone H1 genes

Justin Bodner, Pranathi Vadlamani, Alexander Lee, Kathryn Helmin, Qianli Liu, Almira Pratasenia, Maria Horst, Sudharsana Ravisankar, Sakshi Khurana, Marc Mendillo, Benjamin Singer, Shashank Srivastava, and Daniel Foltz

Corresponding author(s): Daniel Foltz (dfoltz@Northwestern.edu)

Review Timeline:

Submission Date:	17th Jul 25
Editorial Decision:	11th Aug 25
Revision Received:	20th Oct 25
Editorial Decision:	17th Nov 25
Revision Received:	23rd Dec 25
Accepted:	22nd Jan 26

Editor: Esther Schnapp

Transaction Report:

Dear Prof. Foltz,

Thank you for the submission of your manuscript to EMBO reports. We have now received the full set of referee reports that is pasted below.

As you will see, the referees acknowledge that the findings are potentially interesting and add to the recently published papers on the topic. The main concerns relate to the data presentation and interpretation, the methods and the figure legends. Only some more experimental data will need to be provided, i.e. you will need to show when exactly CRAMP-1 is degraded, if and when cells arrest after dTAG treatment, show the TT-RNA Seq data for the 0 and 6 hr points for all histone genes, and provide all data with your ms. However, all referee concerns will need to be addressed and we will need to receive the fully revised ms in mid October in order to publish your paper this year, which is important, given the already published papers on the topic.

Please let me know if you anticipate problems in addressing all concerns and submitting a revised ms by mid October, so that we can discuss this further.

I would thus like to invite you to revise your manuscript with the understanding that the referee concerns must be fully addressed and their suggestions taken on board. Please address all referee concerns in a complete point-by-point response. Acceptance of the manuscript will depend on a positive outcome of a second round of review. It is EMBO reports policy to allow a single round of major revision only and acceptance or rejection of the manuscript will therefore depend on the completeness of your responses included in the next, final version of the manuscript.

You can either publish the study as a short report or as a full article. For short reports, the revised manuscript should not exceed 29,000 characters (including spaces but excluding materials & methods and references) and 5 main plus 5 expanded view figures. The results and discussion sections must further be combined, which will help to shorten the manuscript text by eliminating some redundancy that is inevitable when discussing the same experiments twice. For a normal article there are no length limitations, but it should have more than 5 main figures and the results and discussion sections must be separate. In both cases, the entire materials and methods must be included in the main manuscript file.

- 1) A data availability section providing access to data deposited in public databases is missing. If you have not deposited any data, please add a sentence to the data availability section that explains that.
- 2) Your manuscript contains statistics and error bars based on $n=2$. Please use scatter blots in these cases. No statistics should be calculated if $n=2$.

5) a complete author checklist, which you can download from our author guidelines

<<https://www.embopress.org/page/journal/14693178/authorguide>>. Please insert information in the checklist that is also reflected in the manuscript. The completed author checklist will also be part of the RPF.

6) Please note that all corresponding authors are required to supply an ORCID ID for their name upon submission of a revised manuscript (<<https://orcid.org/>>). Please find instructions on how to link your ORCID ID to your account in our manuscript tracking system in our Author guidelines

<<https://www.embopress.org/page/journal/14693178/authorguide#authorshipguidelines>>

10) Regarding data quantification (see Figure Legends:

<https://www.embopress.org/page/journal/14693178/authorguide#figureformat>)

- the name of the statistical test used to generate error bars and P values,

- the number (n) of independent experiments (please specify technical or biological replicates) underlying each data point,

- the nature of the bars and error bars (s.d., s.e.m.),

- If the data are obtained from n Program fragment delivered error ``Can't locate object method "less" via package "than" (perhaps you forgot to load "than"?) at //ejpvfs23/sites23b/embor_www/letters/embor_decision_revise_and_review.txt line 56.' 2, use scatter blots showing the individual data points.

12) All Materials and Methods need to be described in the main text using our 'Structured Methods' format, which is required for all research articles. According to this format, the Methods section includes a separate Reagents and Tools Table file (listing key reagents, experimental models, software and relevant equipment and including their sources and relevant identifiers) and a Methods and Protocols section describing the methods using a step-by-step protocol format. The aim is to facilitate adoption of the methodologies across labs. More information on how to adhere to this format as well as a downloadable template (.docx) for the Reagents and Tools Table can be found in our author guidelines:

An example of a Method paper with Structured Methods can be found here: <https://www.embopress.org/doi/full/10.1038/s44320-024-00037-6#sec-4>

I look forward to seeing a revised form of your manuscript when it is ready.

Referee #1:

Strengths:

This paper convincingly identifies CRAMP-1 as a critical factor for regulating some histone H1 genes in cultured human cells. Immunoprecipitation of CRAMP-1 identified Gon4L (YARP) as a critical binding partner. CRAMP-1, like Gon4L, is concentrated in the HLB. CRAMP-1 is necessary for high expression of RD-histone mRNAs and at least one histone H1 variant, H1-10. Rapid degradation of CRAMP-1 results in down regulation of histone H1 mRNA as judged by TT-Seq. They have modeled the CRAMP-1/Gon4L interactions by AlphaFold and confirmed some of these interactions by showing interaction between co-expressed proteins. Overall the evidence that CRAMP-1 is involved in expression of histone H1 genes is compelling, and they have identified specific regions of Gon4L and CRAMP-1 that interact and are likely required for this regulation.

Weaknesses:

In many places the writing is confusing, particularly the introduction, where they don't clearly explain what RD-histone genes are [these particular genes and mRNAs are only found in animals (metazoans)] but they cite many yeast papers and refer to higher eucaryotes. The same confusion is found also in presentation of some of the data, some of which is not rigorously supported by the experiments presented, but I think is likely rigorously supported by experiments they did and didn't show. For example, the rapid degradation of CRAMP-1 is a powerful approach, and the most convincing data that this is likely a direct effect is the reduction in H1 expression by TT-Seq by 6 hrs, together with CHIP-Seq showing CRAMP-1 on the promoter of two H1 genes. There are no controls for other histone mRNAs presented for the 6 hr TT-Seq time-point. Critically they don't show when CRAMP-1 is degraded (only show data for 24, 48, 72 hrs). In most cases this method gives degradation in 2 to 4 hrs which is what makes it so powerful.

2. In the RNA Seq data in Fig. 2, which is apparently from 24 hrs after dTAG addition, they find all the RD-histone mRNAs are downregulated (especially H2a, H2b and H3) in two of the replicates after 24 hrs. This is expected if the cells stop growing. An absolute requirement is showing a "simple" flow cytometry to see if and when cells arrest after dTAG treatment, where one can measure both DNA replication (EdU pulse) and DNA content, and whether they are arrested at a particular stage of the cell cycle. They could also wash out the dTAG and see if the cells can resume growth. They should show the TT-RNA Seq for the 0 and 6 hr points for all the histone genes.

Specific Comments:

Introduction:

1. In introduction, no need to mention the gene organization in "model organisms." *Drosophila* has a tandem array with the H2a-H2b gene and H3-H4 gene pairs. Sea urchins has a large repeat with all 5 genes transcribed from the same strand (head to tail). Every organism is different. There are many head to head H2a-H2b gene pairs in mammals, as they mention in the paper.

2. Paragraph two: In metazoans (not higher eucaryotes) RD-histone mRNAs are not polyadenylated but end in a stemloop. In plants they are polyadenylated (no introns in both).

Two references are missing (REF). The 3' UTR does not form a stem-loop. At the end of the 3' UTR there is a conserved stem-loop (6 bp, 4 base loop). CPSF73 is an endonuclease (2 lines from bottom of page 1) that cleaves the nascent transcript to form the 3' end.

Most of the examples in ref.7-11 are from yeast. The RD mRNAs in metazoans are different from all other eucaryotic mRNAs, and the use of yeast examples should be minimized.

3. HLBs are found only in metazoans presumably to provide a subnuclear environment to make the only mRNAs in the cell that are not polyadenylated. The HLB was first named by Joe Gall¹, but not in the reference they cited.

4. On page 3. They should say that the H1-1 to H1-5 genes [and H1-6 gene] all encode non-polyadenylated mRNAs ending in a stem loop and H1-0 and H1-10 are encoded by polyadenylated mRNAs.

Results:

1. The data on Gon4L binding CRAMP-1 and that this is important for regulation of histone H1 expression is convincing, as is the interactions with Gon4L and CRAMP-1 they tested.

2. In general the figure legends are very minimal and need to be expanded. Fig. 1E. They say it is the result of affinity purification [they mean v5-CRAMP-1 IP] of individual H1 genes [they mean H1 proteins!] In the methods for affinity purification they say they used v5 conjugated beads; they must mean anti-v5 conjugated beads. They actually did not do affinity purification, they did immunoprecipitation, which they actually say throughout the methods on top of page 12..

3. While the GON-4L/Clamp-1 interaction is convincing, the finding that H1 proteins are bound by CRAMP-1 is not very convincing. The data in Fig. 4F that NPAT-FLAG binds CRAMP-1 is not at all convincing since they expressed CRAMP-1 and NPAT at high levels in the cell. Unless they get more direct data on these putative interactions on the histone H1/CRAMP-1 interaction, I would delete panels E and F. They don't mention the expression level of v5-Cramp-1 compared to the endogenous CRAMP-1 expression in the v5 tagged line and they need to indicate that. Presumably they have a CRAMP-1 antibody that will let them do that.

4. Looking at the v5 data it looks like more than half of the v5-CRAMP-1 is in the chromatin fraction (Fig. 1G). In panel G, there seems to be one H1 in the nucleoplasm and that protein did IP with CRAMP1). How sure are they that is an H1 protein? The bulk of the H1 migrates more slowly, and does not IP with v5-CLAMP-1 which is expected since the amount of H1 is much larger than the amount of CRAMP-1 in the cell.

5. The RNA data in Fig. 2 is from a 24 hr timepoint which is much too late to see direct effects.

6. Fig. 3 and 4 are fine.

7. Fig. 5, In panel F they show Histone H1-2 and Histone H1-10 and in Fig 5I they show the same data for V5-CRAMP1. Those two panels should be combined

Referee #2:

Bodner et al. examine the relationship between the transcription factor CRAMP1 and histone H1 expression. This is an interesting topic, as it is emerging that individual histone genes or groups of genes, despite clustering together in the HLB and in the genome, experience different regulation. In the case of H1 subtypes, this can influence cell identity and gene expression. The authors discover that CRAMP1 targets a subset of H1 promoters, correlated with promoter methylation, and influences H1 expression. Further, it co-localizes with HLB members, and interacts with GON4L, a negative histone regulator.

Overall, the data are compelling and support the conclusions with the major caveat that the data are poorly explained. Figure legends are extremely minimal, figures are incorrectly referenced (and references are missing) and have confusing labels, and the figures themselves are poorly composed. For example, I have no idea what TT-seq is and it is not explained in the text, figure legend, or Methods. Yet the TT-seq data in Figure 2 could be compelling. The PRC2 data in Figure 2 are not mentioned in text or discussed. There are many other examples (see below). This unfortunately undersells what looks like strong data.

If the data are undersold, the conclusions are slightly oversold. Cramped, the Drosophila homolog of CRAMP1, is already a known HLB factor and specifically regulates H1 expression (PMID: 21336627). This is briefly referenced on page 7, but should be introduced early in the Introduction. Similarly, two other studies have studied CRAMP1 in the context of H1 expression (PMID: 40516528, PMID: 40516529). These are very recent (published this month) and the authors mention the studies in the Discussion, but should bring them up in the introduction as well. Further, these studies have some overlapping discoveries, such as the GON4L interaction and PRC2 relationship. The publications should be compared/contrasted at length in the Discussion.

Major text suggestions:

The first paragraph of the Results is poorly described and impossible for a non-expert to interpret. This is unfortunate, as the corresponding data in Fig 1A-C seem compelling, but they are challenging to interpret. There is little information in the figure legend and none in the Methods.

I have no idea what TT-seq is. It is written at the top of page 6 and in a figure legend. I don't know what it is and it is not explained anywhere in the text. The figure legend is again so minimal that I don't know what the signal is or how to interpret it. Whatever it is, only two histone H1 are shown in the figure- the others should be included in the supplement.

The reference to PRC2 genes in the Discussion is presumably related to Figure 2E, which is impossible to interpret and also not discussed anywhere in the text.

The Figures need to be explicitly referenced in the text. The legends should contain information necessary to interpret the figures.

The authors should spend time in the Discussion comparing what is known about Drosophila Cramped and CRAMP1, as well as the two recent publications (PMID: 40516528, PMID: 40516529). While there is some overlap with the current study, this repeatability strengthens the literature.

The Discussion is currently minimal and much of the Discussion is not particularly relevant. For example, it's unclear why there

is a paragraph that begins "The components of the histone locus body and histone gene expression are strongly conserved across species." This paragraph is more of a list of observations than a discussion of context and sentences do not always follow from the previous thoughts.

Minor suggestions:

Suggest write Results in active voice rather than mixed active/passive voice.

Consider moving CRAMP1 HLB localization (Fig 4A-B) up in the text prior to structural analysis. Maybe even before Fig 2.

"Since CRAMP1 localizes to the HLB" (p5)- the authors have not shown this yet. NPAT is a structural component of the HLB, not a "transcriptional regulator."

Why is NPAT expected to regulate core histone genes (p9)? I believe NPAT regulates all RD histone genes present in the HLB. The authors write that CRAMP1 occupies unmethylated promoters, which, while factual, is misleading. Methylation likely reflects expression status. It's unclear if CRAMP1 is able to occupy these promoters, but does not do so because the genes are not expressed, or if methylation prevents CRAMP1 occupancy.

Where else does CRAMP1 bind in the genome other than the histone locus? If it is binding H1 only at H1 promoters, as well as the negative regulator Mute/GON4L, is it perhaps acting as a feedback mechanism?

H1 transcript levels by RNA-seq or qRT-PCR do not give any indication of regulating transcription, as other factors, such as mRNA turnover, contribute to steady state mRNA levels. There is no evidence that CRAMP1 regulates H1 by transcription. Many figures are incorrectly referenced.

Examples:

Pg 6, bottom: Should reference Fig. 3A instead of 2A

Pg 7, bottom: Should reference Fig. 3D and 3E instead of Fig. 2D and Fig. 2E, respectively.

More incorrectly referenced figures on pages 8 & 9.

Last paragraph of "CRAMP1 localization to Histone Locus Bodies (HLB)" section on page 8 discusses data but lacks any figure references.

Typo: "ChIP-seq experiments have demonstrated that H1 binds almost exclusively to the promoters of histone H1 genes."

Another typo on page 9: "CRAMP1 accumulates at the promoter of the replication coupled H1-2 and H-4 genes..."

Figures:

The figure legends are extremely minimal and not helpful in interpreting the data. The authors should include more information. In addition, sometimes the figures are poorly composed and it is difficult to find the next panel or interpret the order of panels.

Figures are often blurry, have messy text, or are uninterpretable based on what is presented.

Fig 1A is uninterpretable and only minimal information is provided either in text or in the figure legend.

What does "parental" mean in S1 (and in text)?

S2A gives basically no information. What is the reason for the cell picture?

Fig S2 does not show "The addition of dTAG13 resulted in the degradation of CRAMP1 to below detectable levels within 1 d of treatment."

Fig 2B what does "NA" mean? I think it means not expressed (not included in figure legend), but it is misleading to make these dark blue implying downregulation.

Fig 2C: It does seem that there is a relationship between replicate and core histone gene levels. For example, most core genes are downregulated in dTAG replicates 2 and 3. H1 is most compelling, but I think it's misleading to suggest that there is no effect on core histones.

Fig 2F: Please add additional westerns used in quantification in supplement. H3 does look like it is decreased in the western example provided, but not so in the quantification.

Fig 2E is not referenced anywhere in text. Either way, I can't interpret it and the figure legend is no help. This figure is also very messy. I don't know what "PRC2_EED_UP" means but I do know it needs to be replaced with something meaningful and clear.

Fig S2C is not referenced in the text anywhere and is uninterpretable.

Fig S3 requires a lot more explanation in text/in figure legend. SANT domains of WHAT?

S4A/C are unreadably blurry.

Fig 3A: Legend should include that both CRAMP1 and GON4L structures are present in the figure.

Fig 3: Cartoons of the deletion mutants and their domains would be helpful.

Fig 4C is not referenced in the text. What does "NA" mean? It is not in figure legend and is uninterpretable.

Fig 5C is too small to read/interpret.

Why are some ChIP results in Fig 4 and some in Fig 5, and those in Fig 5 discussed before those in Fig 4?

Fig 5D/E: presenting data 1.5 kb in either direction of the histone TSS is not very relevant. Though standard for other genes, the histone genes are so short and clustered that 1.5 kb away from the H4 TSS may also encompass another histone gene. The data LOOK like CRAMP1/NPAT occupy promoters, but in reality the peaks are very broad. Consider shortening the presented range in these data.

Fig 5F: Other H1 genes should be shown in the supplement.

Fig 5G: The legend reads "Distribution of NPAT and CRAMP1 V5 ChIP-seq signal across the gene." I suspect this is actually across all H1 genes and the legend is incredibly misleading.

Fig 5I: Other H1 genes should be shown in the supplement.

Referee #3:

In this manuscript authors discovered a functional association between CRAMP1 and histone H1 through the use of fitness correlations from whole genome KO across multiple cell lines. Then, using a CRAMP1 degron, they describe that CRAMP1 is required for the expression of H1 variants. Then, after AlphaFold predictions, they identify what domains are responsible of the interaction between CRAMP1 and GON4L, a known member of the complex regulating histone biogenesis, and they describe their co-localization to the histone locus body (HLB). Finally, they describe through ChIP-seq the association of CRAMP1 with accessible and demethylated, presumably active H1 variant promoters. Importantly, CRAMP1 may offer specificity for the regulation of H1 variants at the HLB compared to core histone genes.

This manuscript supports two recently-published studies that also discovered the essential involvement of CRAMP1 on histone H1 biogenesis.

It is intriguing an initial observation that CRAMP1 and H1 may interact, presumably at the nucleoplasm, but this was not further investigated.

The study could go deeper in the study of what components of the histone biogenesis complex located at HLB are involved in H1 biogenesis and which factor is responsible of promoter recruitment, as they describe that GON4L recruitment is independent of CRAMP1.

Particular comments are as follow:

1-Improve figure legends with more details, for example Fig.1F & G. Fig S2B, what H1 variant is measured? Fig S2C not mentioned in the text.

2-Fig.1G: CRAMP1-V5 was iPed also from chromatin so it cannot be affirmed that CRAMP1 is not a structural component of the chromatosome. In addition, what are the different bands in the H1 WB? Which one is H1? As abundant in the chromatin input, it seems that upper bands correspond to H1s. What is the band enriched in nucleoplasm, some H1 variant (H1.0)? To be conclusive WB should be performed with H1 variant specific antibodies. Not clear how Fig.S1C relates to this as legends are poor.

3-Contrary to what said in the text, Fig.S2 doesn't show dTAG degradation of CRAMP1 to below detectable levels; this is in Fig.2F. By the way, what H1 variant is being looked at? If pan-H1 antibody, why is there a single band? Fig.2E not mentioned, at least before Fig.2F. Fig.2G not labeled in figures nor mentioned in text (only as legend).

4-Fig.2C it is intriguing the down-regulation of many core histone genes in 2 replicas dTAG. PC analysis of replicates should be shown.

5-Section 'Interaction between CRAMP1-GON4L', revise figure numbering (main and suppl), many mistakes are present. Why Fig.S4 is mentioned before S3...? Etc. S4F should be S4E. Somewhere, 2A should be 3A. Too little proof-reading has been made before submitting...

6-Fig. 4C & D not referred in the text. Fig 4D what are the circles, input and IP, IgG and specific Ab??

7-Pg.8, '...that show discrete localization of CRAMP1 to the HLB (Fig.3A)', should be 4A.
Enlarge text in fig 5C.

8-Fig.5E & F not referred in the text. Fig 5G, what gene(s) are being represented, H1-2+H1-4? Fig 5H not present in the legend nor text: GON4L ChIP-qPCR would be relevant to be reported and discussed. Fig 5H-I-J or I-J-K?
Fig 5J (or I?): What does it means RBBS? H1 variants should be indicated for each line.
I guess the unmethylated H1 promoters might be H1.2, 4 & 10 (CRAMP1 occupied). It coincides with the fact that these 3 variants are unmethylated in all cell lines and universally expressed [<https://doi.org/10.7554/eLife.91306.3>], this could be mentioned.

9-Page 9, 'CRAMP1 is highly recruited to a subset of histone H1 genes, specifically H1-10, H1-2 and H1-4, in RPE1hTERT cells. This is consistent with cell type and tissue dependent expression of histone H1 genes.' H1 variant expression in RPE1hTERT cells should be shown or referenced.
H1 variant gene expression should be tested in RPE1hTERT cells for correlation with ATAC-seq, CRAMP1 and DNA methylation. Are only H1.2, 4 & 10 expressed in RPE cells?
H1-4 does not show ATAC opening? So, what are the expression levels?
H1-10 does not show NPAT, but GON4L? This is relevant and could be discussed.

10-In 293T cells, H1 variants expressed are 0, 2, 3, 4 & 10 (Fig.2B). It would be nice to confirm correlation with CRAMP1 presence (ChIP with HA or endogenous), ATAC and DNA unmethylation (public data probably available for 293T cells).

11-What happens to cell viability upon CRAMP1 degradation long-term (data up to 7 days dTAG is shown)? All expressed H1 variants are being down-regulated. What happens to their promoters in terms of ATAC and DNA methylation? It would be useful to support conclusions in Fig.5.

Are other H1 variants being up-regulated for compensation? This should be shown, at RNA level, or even better at protein level. Also total H1 levels, with Coomassie staining of total H1 extraction or histone extraction, upon CRAMP1 degradation (add to Fig.2F).

If total H1 is considerably decreased, with or without compensation by other H1s, effect on cell viability/cell cycle are expected. It should be tested and discussed.

12-Discussion: 'Our identification of the and as the domains responsible for GON4L binding recapitulates recent findings by Ingham et al.52', complete the sentence.

13-Discussion: '...DNA methylation is the primary mechanism by which histone genes are selected for expression within a cell lineage. We propose further that CRAMP1, either on its own or in a complex, occupies unmethylated H1 promoters to drive expression of H1 genes.' DNA methylation usually is the terminal repressor mechanism triggered by other silencing mechanisms. Here CRAMP1 is suggested as passenger positive factor depending on DNA methylation status of H1 promoters. Investigating consequences on DNA methylation upon CRAMP1 degradation would be useful to clarify mechanisms involved.

14-It has been suggested (Fig.4) that GON4L is upstream CRAMP1 in binding H1 promoters as GON4L binds in the absence of CRAMP1. The opposite could be tested by depleting GON4L. Is GON4L required for CRAMP1 recruitment? A model could be discussed of how recruitment occurs at H1 promoters and what function could exert each factor, also in light of related recent publications.

15-All work is based on tagged CRAMP1 for IP, IF and ChIP. Do you have any proof that tagged protein behaves as the untagged, endogenous CRAMP1? Are there CRAMP1 antibodies that could be used to confirm some of the data?

16-Expression and localization experiments are being made in different cell lines, expressing different H1 variants. As mentioned above, some experiments could be done in the same system, such as measuring H1 expression and CRAMP1 KO effect in RPE1hTERT cells.

17-What could be the sense of H1 and CRAMP1 proteins interacting if confirmed properly?

Referee #4:

The study presents the identification of CRAMP1 as a factor associated to linker histone H1 through CRISPR screens based on cell fitness. The authors then characterize the physical associations of CRAMP1 with other proteins, show CRAMP1 effect on the expression of H1 variants and report its location to the Histone Locus Body (HLB). They also evaluate what domains are key to recruit CRAMP1 to the HLB and to interact with HLB-related factors (i.e. GON4L). They also assess CRAMP1 binding genome-wide and, together with other genome-wide assays, suggest that CRAMP1 is recruited to accessible histone promoters when these are non-methylated.

As noticed by the journal and the authors of this manuscript, Matthews et al., and Ingham et al., Mol Cell, 2025 already described the role of CRAMP1 as a regulator of histone H1 in their publications. Moreover, Matthews et al., also highlights the relationship of CRAMP1 with linker histone H1 and PRC2 through the analysis of DepMap data (Fig 1E of their manuscript). Therefore, the major finding of this manuscript has already been described. However, considering that Bodner et al. is scooping-protected following the rules from EMBO Reports, my assessment will not take these publications into account.

The key finding of Bodner et al. is the discovery of CRAMP1 as a regulator of H1 and its association with the HLB and HLB-related factors. The manuscript is of general interest to the molecular biology community, as it addresses general principles of nuclear function. Overall, their claims are supported by their experiments but there is missing information in the Methods section. It would also be good to include the files resulting from their analysis (further specified below). I would recommend its publication after revision of the comments below.

Major comments

1. The authors have multiple datasets (e.g. CRISPR screen results, AP-MS). Could the authors provide the actual data so that it can be a resource for people to look for specific proteins?
2. In a few instances (e.g. PRC2 components and NPAT), the authors describe specific proteins and their significance in their assays but the dots corresponding to them are not labelled in the Figures. Could the authors please label which dots correspond to them in the CRISPR screen and the AP-MS plots (Fig. 1 and 2) so that one can assess its significance/effect?
3. Regarding the analysis of significance in the CRISPR screens and in the AP-MS, information is missing on the analysis. For

the CRISPR screens, I cannot find a detailed paragraph in the methods section stating what they did exactly and which threshold/cutoff criteria they used for the analysis. For both analyses, while the plot shows color-coded dots, I cannot find the actual threshold values for the log₂FC and corrected p-values that they use. Could they please provide this information in the text/methods/figures?

4. The fact of not finding some proteins as significant across the different datasets could hint towards having a too stringent criteria for significance. E.g. 'NPAT was also found in our AP-MS dataset, it did not meet our cutoff criteria for significant enrichment'. It would be good to know how far from their cutoff criteria NPAT was, as it could be a matter of looking into what threshold would be most suitable for the data in hand.
5. The authors alternate between using RPE1hTERT cells and HEK293 cells. In some cases, there is some reasoning behind why they use one or the other but it is not always clarified. Can the authors state clearly why they use one or the other in each case? It could be confusing for the reader the jump between the two.
6. 'CRAMP-1 degradation led to a reduction in H1 protein as quickly as 1day after the addition of dTAG13' - Figure 2F only shows 1 day as the minimum time explored. Saying 'as quickly as' implies that they would have also checked before 1 day for this specific assay. It could be already degraded before. From the RNA tracks from TT-seq, you can see downregulation already at 6h. Have the authors also checked for CRAMP1 degradation at 6h? This part could also be solved by rephrasing their statement.
7. Regarding the results concerning Fig. 2F. Have the authors assessed or have comments on potential toxicity/off-target effects following degradation?
8. The authors explore the relationship of GON4L and CRAMP1 by checking their interaction through specific domains and by testing whether GON4L binding to promoters is affected by CRAMP1 depletion. With this, they suggest that 'GON4L lies upstream of CRAMP1 recruitment to the HLB and histone H1 promoters' While histone H1 promoters locate to the HLB, the HLB is a structure observed with microscopy. It would be important to provide an image of GON4L staining after depletion of CRAMP1 to support their conclusion.
9. When assessing ATAC-seq, they make the following claim: 'The ATAC-seq analysis shows a correlation between open-chromatin states at the histone H1 genes and the accumulation of CRAMP1 (Fig. 5H,I), suggesting that CRAMP1 is recruited to accessible histone gene promoters.' - at this point, their observations are a correlation and not a functional relationship. An alternative hypothesis is that CRAMP1 could help opening up the chromatin at those points. One way to explore this would be to check ATAC-seq at H1 promoters upon CRAMP1 depletion. However, with the evidence provided, we cannot infer directionality.
10. The authors have these claims:
 - a. 'We observe that DNA methylation negatively correlates with open chromatin and the presence of CRAMP1 at histone gene promoters in general (Fig. 5J). These findings suggest that DNA methylation may be a factor that regulates cell-type specific histone gene expression, and that only non-methylated H1 promoters are accessible to CRAMP1' DNA methylation is known as a repressive mark and is anti-correlated with activity. Their claim is speculative at this stage. They would need functional testing or they would need support from previous literature.
 - b. 'we propose that DNA methylation is the primary mechanism by which histone genes are selected for expression within a cell lineage' this is speculative. It would need further support.

Minor comments

11. 'In addition to the H1 genes, CRAMP1 also showed a high degree of correlation with PRC1 components EED, SUZ12, and EZH2.' These are PRC2 components. Could the authors correct it and also highlight their location in the Figure, as mentioned above?
12. There is a sentence that starts with 'Since CRAMP1 localizes to the HLB' The evidence for it comes later in the paper, so it does not read right. Could you please rephrase?
13. Fig 1G: Can you please add an arrow at the height of the size of the protein?
14. 'Full-length CRAMP1 and the deletion mutant lacking the SANT domain efficiently coprecipitated with GON4L (Fig.2D). However, deletion mutants of CRAMP1 that eliminated the D1 or D2 domains of CRAMP1 abrogated the interaction with GON4L. Likewise, full-length GON4L and the deletion mutant lacking the YY1-binding domain efficiently co-precipitated CRAMP1 (Fig.2E).' should be 3D and 3E
15. 'Alternatively, deletion of either the D1 or D2 CRAMP1 domain resulted in a protein that was unable to localize to the HLB.' I would say concentrate instead of 'localize'. It may localize a bit but not enriched.

16. ChIP-seq was conducted using the RPE1hTERT CRAMP1-V5 expressing cells that show discrete localization of CRAMP1 to the HLB (Fig. 3A). This is probably figure 4.

17. Some references are missing - it says REF

18. 'Our identification of the and as the domains responsible for GON4L binding recapitulates recent findings by Ingham et al.52 and goes on to demonstrate that the PAH repeats of GON4L provide the interface for CRAMP1 binding' -> this sentence needs rephrasing ('the and').

Suggestion

It is nice to see the interaction of GON4L and CRAMP1 at multiple levels: physical interaction, protein domains (AlphaFold + depletions + co-IPs), HLB colocalization. However, I find two aspects missing that would round up their characterization and support their conclusions:

1. They do not show the effect of GON4L enrichment at the HLB by imaging upon the depletion of specific domains (4C). It would be a nice addition to have GON4L staining in 4C, as it would round up their characterization and support their conclusions.
2. If they have the data, they could show GON4L ChIP-seq in relation to CRAMP1 and NPAT.

We appreciate the helpful comments that were provided by the reviewers that have significantly improved the text of the manuscript and the impact of the work. We hope that the changes that have been made to the manuscript will largely satisfy the reviewers concerns. We also apologize that we have not been able to pursue several excellent suggestions due to the limitations of the review process. But look forward to continuing our experiments to understand the function of CRAMP1.

Referee #1:

Strengths:

This paper convincingly identifies CRAMP-1 as a critical factor for regulating some histone H1 genes in cultured human cells. Immunoprecipitation of CRAMP-1 identified Gon4L (YARP) as a critical binding partner. CRAMP-1, like Gon4L, is concentrated in the HLB. CRAMP-1 is necessary for high expression of RD-histone mRNAs and at least one histone H1 variant, H1-10. Rapid degradation of CRAMP-1 results in down regulation of histone H1 mRNA as judged by TT-Seq. They have modeled the CRAMP-1/Gon4L interactions by AlphaFold and confirmed some of these interactions by showing interaction between co-expressed proteins. Overall the evidence that CRAMP-1 is involved in expression of histone H1 genes is compelling, and they have identified specific regions of Gon4L and CRAMP-1 that interact and are likely required for this regulation.

We thank the reviewer for their positive comments and assessment of the manuscript!

Weaknesses:

In many places the writing is confusing, particularly the introduction, where they don't clearly explain what RD-histone genes are [these particular genes and mRNAs are only found in animals (metazoans)] but they cite many yeast papers and refer to higher eucaryotes. The same confusion is found also in presentation of some of the data, some of which is not rigorously supported by the experiments presented, but I think is likely rigorously supported by experiments they did and didn't show.

We appreciate the comments and have worked to make the manuscript more clearly written and detailed.

For example, the rapid degradation of CRAMP-1 is a powerful approach, and the most convincing data that this is likely a direct effect is the reduction in H1 expression by TT-Seq by 6 hrs, together with CHIP-Seq showing CRAMP-1 on the promoter of two H1 genes. There are no controls for other histone mRNAs presented for the 6 hr TT-Seq time-point .

Critically they don't show when CRAMP-1 is degraded (only show data for 24,48, 72 hrs). In most cases this method gives degradation in 2 to 4 hrs which is what makes it so powerful.

We thank the reviewer for pointing this out. We did, in fact, test the degradation after acute dTAG treatment, but neglected to include it in the original manuscript. The data are included in figure 2H of the revised manuscript. As suggested by the reviewer, the degradation of CRAMP1 occurs within 3-5 hours after the addition of dTAG1. Therefore, the TT-seq assays are conducted under conditions where steady-state levels of the CRAMP1 protein are less than 10%. In addition, the TT-seq analysis includes a complete dataset showing the effect of CRAMP1 depletion on core histones and a random set of expressed genes versus the H1 linker histone in figure 2J. In addition, the normalized count levels for each expressed histone gene are included in the figure EV2F.

2. In the RNA Seq data in Fig. 2, which is apparently from 24 hrs after dTAG addition, they find all the RD-histone mRNAs are downregulated (especially H2a, H2b and H3) in two of the replicates after 24 hrs. This is expected if the cells stop growing. An absolute requirement is showing a "simple" flow cytometry to see if and when cells arrest after dTAG treatment, where one can measure both DNA replication (EdU pulse) and DNA content, and whether they are arrested at a particular stage of the cell cycle. They could also wash out the dTAG and see if the cells can resume growth

CRAMP1 is not an essential gene in human cell culture. We provide growth curves of HEK293T^{CRAMP1-HA-dTAG} cells treated with dTAG for up to 6 days and these cells show no growth difference (Fig.2B). Likewise, viability from the DepMap (depmap.org) shows that CRAMP1 knockout has only a mild effect on cell viability over longer time frames across many different cell types and is not considered essential. Therefore, all the phenotypes that we observed in the manuscript are not due to a slowing of the cell cycle or decreased cell viability. Data from the Ingham et al (2025) indicate that CRAMP1 does become essential when TopII is inhibited. This is specifically mentioned in the discussion of the revised manuscript.

They should show the TT-RNA Seq for the 0 and 6 hr points for all the histone genes.

As noted above in point 1, read counts for each of the core histone and linker histone genes are displayed in figure EV2F. Only the linker histones, as a group, are consistently down in the 6- and 24-hour timepoints. Various core histones do show some reduction, especially H2A variants, but the degree of reduction is generally much less than what is observed in the expression of histone H1 genes. This is now discussed in text of the manuscript.

Specific Comments:

Introduction:

1. In introduction, no need to mention the gene organization in "model organisms." *Drosophila* has a tandem array with the H2a-H2b gene and H3-H4 gene pairs. Sea urchins has a large repeat with all 5 genes transcribed from the same strand (head to tail). Every organism is different. There are many head to head H2a-H2b gene pairs in mammals, as they mention in the paper.

We have removed the discussion of the arrangement of histone genes from the introduction.

2. Paragraph two: In metazoans (not higher eucaryotes) RD-histone mRNAs are not polyadenylated but end in a stemloop. In plants they are polyadenylated (no introns in both).

Two references are missing (REF). The 3' UTR does not form a stem-loop. At the end of the 3' UTR there is a conserved stem-loop (6 bp, 4 base loop). CPSF73 is an endonuclease (2 lines from bottom of page 1) that cleaves the nascent transcript to form the 3' end.

Most of the examples in ref.7-11 are from yeast. The RD mRNAs in metazoans are different from all other eucaryotic mRNAs, and the use of yeast examples should be minimized.

These references have been corrected in the revised manuscript.

3. HLBs are found only in metazoans presumably to provide a subnuclear environment to make the only mRNAs in the cell that are not polyadenylated. The HLB was first named by Joe Gall¹, but not in the reference they cited.

We have cited Dr. Gall's seminal work in the revised manuscript.

4. On page 3. They should say that the H1-1 to H1-5 genes [and H1-6 gene] all encode non-polyadenylated mRNAs ending in a stem loop and H1-0 and H1-10 are encoded by polyadenylated mRNAs.

These points have been addressed in the revised introduction of the manuscript.

Results:

1. The data on Gon4L binding CRAMP-1 and that this is important for regulation of histone H1 expression is convincing, as is the interactions with Gon4L and CRAMP-1 they tested.

We thank the reviewer for the positive comments.

2. In general the figure legends are very minimal and need to be expanded. Fig. 1E. They say it is the result of affinity purification [they mean v5-CRAMP-1 IP] of individual H1 genes [they mean H1 proteins!] In the methods for affinity purification they say they used v5 conjugated beads; they must mean anti-v5 conjugated beads. They actually did not do affinity purification, they did immunoprecipitation, which they actually say throughout the methods on top of page 12..

In general, we have expanded the amount of information available in the figure legends of the revised manuscript. With respect to the figure legend for figure 1, we have clarified the information for panel 1E to read that these values were extracted from the CRAMP1-V5 immunoprecipitation in panel D. Although IP-MS is generally an affinity purification technique, we have changed the text of the manuscript to more clearly describe the V5-immunoprecipitation approach we employed.

3. While the GON-4L/Clamp-1 interaction is convincing, the finding that H1 proteins are bound by CRAMP-1 is not very convincing. The data in Fig. 4F that NPAT-FLAG binds CRAMP-1 is not at all convincing since they expressed CRAMP-1 and NPAT at high levels in the cell. Unless they get more direct data on these putative interactions on the histone H1/CRAMP-1 interaction, I would delete panels E and F. They don't mention the expression level of v5-Cramp-1 compared to the endogenous CRAMP-1 expression in the v5 tagged line and they need to indicate that. Presumably they have a CRAMP-1 antibody that will let them do that.

We agree that the data in panel E and F do not demonstrate a direct interaction between CRAMP1 and NPAT or histone H1. Nor do we propose that is the case. The interaction with NPAT is more exhaustively explored in the manuscript from Matthews et al (PMID: PMID: 40516528). Therefore, we have removed the NPAT co-IP from our dataset, since it was limited in scope. However, we are keen to retain the data suggesting that CRAMP1 is associated with histone H1 (direct or indirect) because we think this observation will be important in understanding CRAMP1 function in the future.

Unfortunately, we have not identified an antibody that provides reliable signal to detect endogenous CRAMP1 by immunoblot, so we are unable to determine the relative expression of endogenous CRAMP1 versus CRAMP1-V5. However, the functional experiments in the manuscript were conducted using the CRAMP1-HA-dTAG cells line that express the tagged protein from the endogenous locus.

4. Looking at the v5 data it looks like more than half of the v5-CRAMP-1 is in the

chromatin fraction (Fig. 1G). In panel G, there seems to be one H1 in the nucleoplasm and that protein did IP with CRAMP1). How sure are they that is an H1 protein? The bulk of the H1 migrates more slowly, and does not IP with v5-CLAMP-1 which is expected since the amount of H1 is much larger than the amount of CRAMP-1 in the cell.

In Figure 1G, we expect that the lower band is the H1 specific band, based its expected molecular weight. The antibody was raised against H1-0 and has been shown previously to recognize a band of ~ 30kD. However, we cannot exclude that the slower migrating band is not a modified version of H1 (or a different variant). The lower band has been noted in the revised manuscript as the most likely H1-0 band. Either way, the data support our hypothesis that H1-0 is bound primarily to CRAMP1 in the soluble fraction.

5. The RNA data in Fig. 2 is from a 24 hr timepoint which is much too late to see direct effects.

The data in Figure 2 include both steady state RNA-seq data and TT-seq to detect nascent gene transcription. We agree that the 24hr timepoint is not sufficient to determine that the effect is directly on H1 transcription. However, the 6 hour timepoint in the TT-seq experiments (Figure 2G-I) clearly show that nascent histone H1 gene expression is consistently and significantly reduced by loss of CRAMP1.

6. Fig. 3 and 4 are fine.

7. Fig. 5, In panel F they show Histone H1-2 and Histone H1-10 and in Fig 5I they show the same data for V5-CRAMP1. Those two panels should be combined

We have combined the track example in Figure 5F and I in the original manuscript into Figure 5F of the revised manuscript.

Referee #2:

Bodner et al. examine the relationship between the transcription factor CRAMP1 and histone H1 expression. This is an interesting topic, as it is emerging that individual histone genes or groups of genes, despite clustering together in the HLB and in the genome, experience different regulation. In the case of H1 subtypes, this can influence cell identity and gene expression. The authors discover that CRAMP1 targets a subset of H1 promoters, correlated with promoter methylation, and influences H1 expression.

Further, it co-localizes with HLB members, and interacts with GON4L, a negative histone regulator.

Overall, the data are compelling and support the conclusions with the major caveat that the data are poorly explained. Figure legends are extremely minimal, figures are incorrectly referenced (and references are missing) and have confusing labels, and the figures themselves are poorly composed. For example, I have no idea what TT-seq is and it is not explained in the text, figure legend, or Methods. Yet the TT-seq data in Figure 2 could be compelling. The PRC2 data in Figure 2 are not mentioned in text or discussed. There are many other examples (see below). This unfortunately undersells what looks like strong data.

We appreciate the reviewer's suggestions for improving the quality of the manuscript. The revision includes expanded explanations in the figure legends, includes a paragraph explaining the rationale for the use of TT-seq and more fully describes the data. In addition, the PRC2 data are explicitly referenced and described in the text of the manuscript.

If the data are undersold, the conclusions are slightly oversold. Cramped, the *Drosophila* homolog of CRAMP1, is already a known HLB factor and specifically regulates H1 expression (PMID: 21336627). This is briefly referenced on page 7, but should be introduced early in the Introduction. Similarly, two other studies have studied CRAMP1 in the context of H1 expression (PMID: 40516528, PMID: 40516529). These are very recent (published this month) and the authors mention the studies in the Discussion, but should bring them up in the introduction as well. Further, these studies have some overlapping discoveries, such as the GON4L interaction and PRC2 relationship. The publications should be compared/contrasted at length in the Discussion.

We have extended our discussion of the recently published manuscripts (PMID: 40516528, PMID: 40516529).

Major text suggestions:

The first paragraph of the Results is poorly described and impossible for a non-expert to interpret. This is unfortunate, as the corresponding data in Fig 1A-C seem compelling, but they are challenging to interpret. There is little information in the figure legend and none in the Methods.

We have added additional details to the first paragraph of the results section to describe the Fitness correlations and added information about the approach in the materials and methods.

I have no idea what TT-seq is. It is written at the top of page 6 and in a figure legend. I don't know what it is and it is not explained anywhere in the text. The figure legend is again so minimal that I don't know what the signal is or how to interpret it. Whatever it is, only two histone H1 are shown in the figure- the others should be included in the supplement.

The revised manuscript includes a fuller dataset that includes the TT-seq signal for all histones in aggregate and individually. These data are included in Figure 2I and supplemental figure S2F. A rationale is provided for using the TT-seq approach and the text provides a reference for the technique.

The reference to PRC2 genes in the Discussion is presumably related to Figure 2E, which is impossible to interpret and also not discussed anywhere in the text.

The PRC2 data are now specifically referenced in the results section of the manuscript. In addition, we have included the volcano plot of the RNA-seq data after 7D of dTAG treatment in figure EV2E.

The Figures need to be explicitly referenced in the text. The legends should contain information necessary to interpret the figures.

Figure legends have been expanded to be more descriptive.

The authors should spend time in the Discussion comparing what is known about *Drosophila Cramped* and CRAMP1, as well as the two recent publications (PMID: 40516528, PMID: 40516529). While there is some overlap with the current study, this repeatability strengthens the literature.

The discussion has been expanded to include the *Drosophila Cramped* protein and the recent publications.

The Discussion is currently minimal and much of the Discussion is not particularly relevant. For example, it's unclear why there is a paragraph that begins "The components of the histone locus body and histone gene expression are strongly conserved across species." This paragraph is more of a list of observations than a discussion of context and sentences do not always follow from the previous thoughts.

We have edited the discussion to make it more relevant to the current work and hopefully flow more logically.

Minor suggestions:

Suggest write Results in active voice rather than mixed active/passive voice.

We edited the results section to consistently use the active voice.

Consider moving CRAMP1 HLB localization (Fig 4A-B) up in the text prior to structural analysis. Maybe even before Fig 2.

We appreciate the reviewer's suggestion. However, we thought it was best to keep the microscopy-based localization of CRAMP1, and the mutants that disrupt its localization, in the same figure (Fig.4). In addition, we felt this was best placed immediately after the mutants we identified in figure 3.

"Since CRAMP1 localizes to the HLB" (p5)- the authors have not shown this yet. NPAT is a structural component of the HLB, not a "transcriptional regulator."

Why is NPAT expected to regulate core histone genes (p9)? I believe NPAT regulates all RD histone genes present in the HLB.

We have edited the text on page 5 to ensure that it coincides with the flow of the manuscript. And we edited the description of NPAT function on pages 5 and 9

The authors write that CRAMP1 occupies unmethylated promoters, which, while factual, is misleading. Methylation likely reflects expression status. It's unclear if CRAMP1 is able to occupy these promoters, but does not do so because the genes are not expressed, or if methylation prevents CRAMP1 occupancy.

We have changed this text to read "CRAMP1 occupies the promoters of actively expressed replication-coupled linker histone genes....."

Where else does CRAMP1 bind in the genome other than the histone locus? If it is binding H1 only at H1 promoters, as well as the negative regulator Mute/GON4L, is it perhaps acting as a feedback mechanism?

The high confidence binding sites lie exclusively within the histone H1 gene promoters. We agree that there may be more complex regulatory process that includes GON4L and perhaps H1 binding to CRAMP1. We look forward to dissecting the pathway in future work.

H1 transcript levels by RNA-seq or qRT-PCR do not give any indication of regulating transcription, as other factors, such as mRNA turnover, contribute to steady state mRNA levels. There is no evidence that CRAMP1 regulates H1 by transcription.

We agree that RNA-seq and qRT-PCR are a measure of steady-state RNA levels. However, we also provide data using TT-seq to show that nascent transcription is altered within 6 hours of CRAMP1 degradation. In addition, we show that CRAMP1 occupies the promoter of the H1 genes, that also show changes in steady state and nascent transcript levels. Together, we feel this is a compelling case that CRAMP1 regulates the transcription of histone H1 genes.

Many figures are incorrectly referenced.

Examples:

Pg 6, bottom: Should reference Fig. 3A instead of 2A

Pg 7, bottom: Should reference Fig. 3D and 3E instead of Fig. 2D and Fig. 2E, respectively.

More incorrectly referenced figures on pages 8 & 9.

Last paragraph of "CRAMP1 localization to Histone Locus Bodies (HLB)" section on page 8 discusses data but lacks any figure references.

Typo: "ChIP-seq experiments have demonstrated that H1 binds almost exclusively to the promoters of histone H1 genes."

Another typo on page 9: "CRAMP1 accumulates at the promoter of the replication coupled H1-2 and H-4 genes..."

The typos and incorrect references have been corrected.

Figures:

The figure legends are extremely minimal and not helpful in interpreting the data. The authors should include more information. In addition, sometimes the figures are poorly composed and it is difficult to find the next panel or interpret the order of panels. Figures are often blurry, have messy text, or are uninterpretable based on what is presented. Fig 1A is uninterpretable and only minimal information is provided either in text or in the figure legend.

We have added details to both the text and figure legends to better describe the approach in figure 1. Likewise we have ensured that quality and flow of the remaining figures.

What does "parental" mean in S1 (and in text)?

“Parental” refers to the cell that was used to create the stable cell lines. In each case we have clarified that these are HEK293T cells.

S2A gives basically no information. What is the reason for the cell picture?

The figure in S2A has been replaced in the revised manuscript with a diagram of the strategy for CRISPR-Cas9 endogenous tagging of the CRAMP1 c-terminus with the HA-dTAG.

Fig S2 does not show "The addition of dTAG13 resulted in the degradation of CRAMP1 to below detectable levels within 1 d of treatment."

The appropriate figure reference has been added to be Fig. 2A.

Fig 2B what does "NA" mean? I think it means not expressed (not included in figure legend), but it is misleading to make these dark blue implying downregulation.

Figure 2E (which was figure 2B in the original manuscript) shows overall expression level in unperturbed cells. NA means that it is not identified in the RNA-seq sample, suggesting it is not expression. We have updated the figure to include N.D. = not detected.

Fig 2C: It does seem that there is a relationship between replicate and core histone gene levels. For example, most core genes are downregulated in dTAG replicates 2 and 3. H1 is most compelling, but I think it's misleading to suggest that there is no effect on core histones.

We agree that there are variations in the core histones, especially in H2A, that indicate that CRAMP1 may have a broader effect on regulation of the HLB. We have altered the text to acknowledge the changes that are present. However, we also show that CRAMP1 loss induces a much larger change in H1 genes than it does in core histone gene expression. Combined with the localization of CRAMP1 to H1 promoters we think the changes in H2A may be secondary.

Fig 2F: Please add additional westerns used in quantification in supplement. H3 does look like it is decreased in the western example provided, but not so in the quantification.

Below we display all three replicates used for quantification of H1 and H3 following long term CRAMP1 degradation in figure 2C. While there is some variability in the H3 levels, there is no significant reduction across replicates. These source data are included with the revised manuscript.

Reviewer Figure 1 Three replicates used in figure 2C to quantify the reduction of histone H1 and H3 in response to CRAMP1 degradation.

Fig 2E is not referenced anywhere in text. Either way, I can't interpret it and the figure legend is no help. This figure is also very messy. I don't know what "PRC2_EED_UP" means but I do know it needs to be replaced with something meaningful and clear.

In the revised manuscript, we discuss Figure 2E (which is 2G in the revised manuscript) and the results section and have re-labeled the figure for clarity.

Fig S2C is not referenced in the text anywhere and is uninterpretable.

Figure S2C is now referenced on Page 6 of the revised manuscript.

Fig S3 requires a lot more explanation in text/in figure legend. SANT domains of WHAT?

The figure legends have been expanded and the figure modified to provide additional information.

S4A/C are unreadably blurry.

We have assured that the images provided in the submitted manuscript are high quality and should yield readable results.

Fig 3A: Legend should include that both CRAMP1 and GON4L structures are present in the figure.

We have modified the figure legend for Fig.3C (formerly figure 3A) to explicitly state that the structure includes CRAMP1 in complex with the 2xPAH domains of GON4L. Domains are color coded consistent with panels A and B.

Fig 3: Cartoons of the deletion mutants and their domains would be helpful.

Cartoons depicting the deletion mutants has been added to figure 3D and E.

Fig 4C is not referenced in the text. What does "NA" mean? It is not in figure legend and is uninterpretable.

The images in 4C have been relabeled, NA has been replaced with "No addition".

Fig 5C is too small to read/interpret.

Text has been enlarged.

Why are some ChIP results in Fig 4 and some in Fig 5, and those in Fig 5 discussed before those in Fig 4?

In the revised manuscript we have moved the ChIP-PCR experiments originally presented in figure 4, to figure 5. This consolidates the ChIP experiments into figure 5.

Fig 5D/E: presenting data 1.5 kb in either direction of the histone TSS is not very relevant. Though standard for other genes, the histone genes are so short and clustered that 1.5 kb away from the H4 TSS may also encompass another histone gene. The data LOOK like CRAMP1/NPAT occupy promoters, but in reality, the peaks are very broad. Consider shortening the presented range in these data.

While the issue of spacing between histone genes is relevant for the core histones, the TSS of H1 genes are farther than 1.5 KB away from any neighboring histone gene TSS (core or linker). So the NPAT localization that we observe for H1-2 and H1-4 genes is bone fide localization to the H1 linker gene TSS.

Fig 5F: Other H1 genes should be shown in the supplement.

The occupancy of and ATAC seq data for all H1 genes are shown in figure 5E and 5H.

Fig 5G: The legend reads "Distribution of NPAT and CRAMP1 V5 ChIP-seq signal across the gene." I suspect this is actually across all H1 genes and the legend is incredibly misleading.

Figure 5G has been removed from the revised manuscript.

Fig 5I: Other H1 genes should be shown in the supplement.

The ATAC-seq signal is shown for all H1 genes in revised 5H.

Referee #3:

In this manuscript authors discovered a functional association between CRAMP1 and histone H1 through the use of fitness correlations from whole genome KO's across multiple cell lines. Then, using a CRAMP1 degron, they describe that CRAMP1 is required for the expression of H1 variants. Then, after AlphaFold predictions, they identify what domains are responsible of the interaction between CRAMP1 and GON4L, a known member of the complex regulating histone biogenesis, and they describe their co-localization to the histone locus body (HLB). Finally, they describe through ChIP-seq the association of CRAMP1 with accessible and demethylated, presumably active H1 variant promoters. Importantly, CRAMP1 may offer specificity for the regulation of H1 variants at the HLB compared to core histone genes.

This manuscript supports two recently-published studies that also discovered the essential involvement of CRAMP1 on histone H1 biogenesis.

It is intriguing an initial observation that CRAMP1 and H1 may interact, presumably at the nucleoplasm, but this was not further investigated.

The study could go deeper in the study of what components of the histone biogenesis complex located at HLB are involved in H1 biogenesis and which factor is responsible of promoter recruitment, as they describe that GON4L recruitment is independent of CRAMP1.

Particular comments are as follow:

1-Improve figure legends with more details, for example Fig.1F & G. Fig S2B, what H1 variant is measured? Fig S2C not mentioned in the text.

Figure legends have been expanded in include that variant is H1-0. And the genes that are assayed in figure EV2 are specifically referenced in the figure. Figure S2C (now EV2D) is now specifically referenced in the text of the manuscript.

2-Fig.1G: CRAMP1-V5 was iPed also from chromatin so it cannot be affirmed that CRAMP1 is not a structural component of the chromatosome. In addition, what are the different bands in the H1 WB? Which one is H1? As abundant in the chromatin input, it seems that upper bands correspond to H1s. What is the band enriched in nucleoplasm, some H1 variant (H1.0)? To be conclusive WB should be performed with H1 variant specific antibodies. Not clear how Fig.S1C relates to this as legends are poor.

(copied for comment to reviewer #1) In Figure 1G, we expect that the lower band is the H1 specific band, based its expected molecular weight. The antibody was raised against H1-0 and has been shown previously to recognize a band of ~ 30kD. However, we cannot exclude that the slower migrating band is not a modified version of H1 (or a different variant). The lower band has been noted in the revised manuscript as the most likely H1-0 band. Either way, the data support our hypothesis that H1-0 is bound primarily to CRAMP1 in the soluble fraction.

3-Contrary to what said in the text, Fig.S2 doesn't show dTAG degradation of CRAMP1 to below detectable levels; this is in Fig.2F. By the way, what H1 variant is being looked at? If pan-H1 antibody, why is there a single band?

The antibody is raised against Histone H1-0. This is noted in the figure legend and the material and methods of the revised manuscript.

Fig.2E not mentioned, at least before Fig.2F. Fig.2G not labeled in figures nor mentioned in text (only as legend).

The legend and labels for figure 2 have been updated and are referenced in the text and legend.

4-Fig.2C it is intriguing the down-regulation of many core histone genes in 2 replicas dTAG. PC analysis of replicates should be shown.

(copied from comment to reviewer #2) - We agree that there are variations in the core histones, especially in H2A, that indicate that CRAMP1 may have a broader effect on regulation of the HLB. We have altered the text to acknowledge the changes that are present. We also show that CRAMP1 loss induces a much larger change in H1 genes than it does in core histone gene expression.

5-Section 'Interaction between CRAMP1-GON4L', revise figure numbering (main and suppl), many mistakes are present. Why Fig.S4 is mentioned before S3...? Etc. S4F should be S4E. Somewhere, 2A should be 3A. Too little proof-reading has been made before submitting...

Figure legends were updated and verified prior to resubmission of the manuscript.

6-Fig. 4C & D not referred in the text. Fig 4D what are the circles, input and IP, IgG and specific Ab??

Figures 4C and 4D are specifically referenced in the revised manuscript. The circles in Figure 4D are defined in the figure and in the figure legend.

7-Pg.8, '...that show discrete localization of CRAMP1 to the HLB (Fig.3A)', should be 4A.

Enlarge text in fig 5C.

Reference to figure 4A has been corrected and text has been enlarged in figure 5C.

8-Fig.5E & F not referred in the text. Fig 5G, what gene(s) are being represented, H1-2+H1-4? Fig 5H not present in the legend nor text: GON4L ChIP-qPCR would be relevant to be reported and discussed. Fig 5H-I-J or I-J-K?

Fig. 5E and F are now referenced in the revised manuscript. Figure 5G has been removed because it does not provide significant insight and is not discussed in the manuscript, and Fig.5H has been removed because it is redundant with figure 4D.

Fig 5J (or I?): What does it mean RBBS? H1 variants should be indicated for each line. I guess the unmethylated H1 promoters might be H1.2, 4 & 10 (CRAMP1 occupied).

RRBS (reduced representation bisulfite sequencing) is now defined in the text and the figure legend of Fig 5. In addition, we have added labels to the figure to indicate the corresponding H1 variants that correspond to the individual rows.

It coincides with the fact that these 3 variants are unmethylated in all cell lines and universally expressed [<https://doi.org/10.7554/eLife.91306.3>], this could be mentioned. 9-Page 9, 'CRAMP1 is highly recruited to a subset of histone H1 genes, specifically H1-10, H1-2 and H1-4, in RPE1hTERT cells. This is consistent with cell type and tissue dependent expression of histone H1 genes.' H1 variant expression in RPE1hTERT cells should be shown or referenced.

We have added the Salinas-Pena et al. reference to the revised manuscript.

H1 variant gene expression should be tested in RPE1hTERT cells for correlation with ATAC-seq, CRAMP1 and DNA methylation. Are only H1.2, 4 & 10 expressed in RPE cells?

H1-4 does not show ATAC opening? So, what are the expression levels?

We were unable to complete this analysis in the timeframe required for resubmission of the manuscript. In other cell types, the ATAC-seq signals accurately reflect the

expression pattern of the variant, and so we are confident that the ATAC-seq is a good proxy for RNA expression.

H1-10 does not show NPAT, but GON4L? This is relevant and could be discussed.

We have added a discussion of the GON4L specificity to H1 promoters independent of replication coupling of the gene.

10-In 293T cells, H1 variants expressed are 0, 2, 3, 4 & 10 (Fig.2B). It would be nice to confirm correlation with CRAMP1 presence (ChIP with HA or endogenous), ATAC and DNA unmethylation (public data probably available for 293T cells).

We demonstrated the correlation between recruitment of CRAMP1 (by V5-Chip), opening of chromatin and the DNA methylation in the same RPE1hTERT cells lines in figure 5. We were concerned that adding publicly available HEK293T data would not add significantly to this results. Especially because any discrepancies between our expression data and the public ATAC/DNA methylation data could be due to cell lines differences.

11-What happens to cell viability upon CRAMP1 degradation long-term (data up to 7 days dTAG is shown)? All expressed H1 variants are being down-regulated. What happens to their promoters in terms of ATAC and DNA methylation? It would be useful to support conclusions in Fig.5.

Are other H1 variants being up-regulated for compensation? This should be shown, at RNA level, or even better at protein level.

While we agree that the questions posed about how cells respond to H1 loss over time are important, however, we were unable to complete additional ATAC and/or DNA methylation analysis at longer time points within the timeframe required for the resubmission of the manuscript.

Also total H1 levels, with Coomassie staining of total H1 extraction or histone extraction, upon CRAMP1 degradation (add to Fig.2F).

We provide RNA-seq, rtPCR and immunoblot evidence from whole-cell extracts to demonstrate that CRAMP1 is a positive regulator of histone H1 gene expression. We feel that the whole-cell extract is the best approach to ensure that if changes in protein stability occur preferentially in the soluble or chromatin fraction, that we would be sure to observe them.

If total H1 is considerably decreased, with or without compensation by other H1s, effect on cell viability/cell cycle are expected. It should be tested and discussed.

(From comments to reviewer 1): CRAMP1 is not an essential gene in human cell culture. We provide growth curves of HEK293T^{CRAMP1-HA-dTAG} cells treated with dTAG for up to 6 days and these cells show no growth difference. Likewise, viability from the DepMAP (depmap.org) shows that CRAMP1 knockout has only mild effects on cell viability over longer time frames across many different cell types and is not considered essential. Therefore, all the phenotypes that we observed in the manuscript are not due to a slowing of the cell cycle or decreased cell viability. Data from the Ingham et al (2025) indicate that CRAMP1 does become essential when TopII is inhibited. This is specifically mentioned in the discussion of the revised manuscript.

12-Discussion: 'Our identification of the and as the domains responsible for GON4L binding recapitulates recent findings by Ingham et al.⁵²', complete the sentence.

We corrected the sentence in the revised manuscript.

13-Discussion: '...DNA methylation is the primary mechanism by which histone genes are selected for expression within a cell lineage. We propose further that CRAMP1, either on its own or in a complex, occupies unmethylated H1 promoters to drive expression of H1 genes.' DNA methylation usually is the terminal repressor mechanism triggered by other silencing mechanisms. Here CRAMP1 is suggested as passenger positive factor depending on DNA methylation status of H1 promoters. Investigating consequences on DNA methylation upon CRAMP1 degradation would be useful to clarify mechanisms involved.

We agree that understanding how CRAMP1, GON4L and other regulators of histone expression affect the methylation status of histones gene promoters to determine the primary determinants of histone gene regulation. However, we feel this will require significant additional experimental efforts that are beyond the scope of the current manuscript.

14-It has been suggested (Fig.4) that GON4L is upstream CRAMP1 in binding H1 promoters as GON4L binds in the absence of CRAMP1. The opposite could be tested by depleting GON4L. Is GON4L required for CRAMP1 recruitment? A model could be discussed of how recruitment occurs at H1 promoters and what function could exert each factor, also in light of related recent publications.

We are anxious to test the effect of GON4L loss on CRAMP1 requirement. However, to date, we have not achieved cell lines containing the endogenously tagged GON4L

containing the degron dTAG. Recently published work is discussed in the revised manuscript.

15-All work is based on tagged CRAMP1 for IP, IF and ChIP. Do you have any proof that tagged protein behaves as the un-tagged, endogenous CRAMP1? Are there CRAMP1 antibodies that could be used to confirm some of the data?

In the revised manuscript, we show that endogenous CRAMP1 colocalizes with the HLB protein Symplekin (Fig. EV4G) demonstrating that both the endogenously tagged and endogenous proteins are localized to the HLB.

16-Expression and localization experiments are being made in different cell lines, expressing different H1 variants. As mentioned above, some experiments could be done in the same system, such as measuring H1 expression and CRAMP1 KO effect in RPE1hTERT cells.

We have validated that CRAMP1 suppression in RPE1-hTERT cells by siRNA (Fig. EV2C,D) or by dTAG mediated destruction in HEK293T cells (Fig. 2), suggesting that the process is conserved across cell types. The timeframe limits our ability to create CRAMP1 KO in RPE1hTERT cells.

17-What could be the sense of H1 and CRAMP1 proteins interacting if confirmed properly?

We expect that the interaction of CRAMP1 with both the H1 gene promoter and the H1 protein may represent a feedback mechanism that is sensitive to the protein level so H1. Alternatively, CRAMP1 may act as a chaperone for H1 stability, such that chaperone activity is ensured for H1 that is expressed in the cell. These two potential models will be tested in future experiments.

Referee #4:

The study presents the identification of CRAMP1 as a factor associated to linker histone H1 through CRISPR screens based on cell fitness. The authors then characterize the physical associations of CRAMP1 with other proteins, show CRAMP1 effect on the expression of H1 variants and report its location to the Histone Locus Body (HLB). They also evaluate what domains are key to recruit CRAMP1 to the HLB and to interact with HLB-related factors (i.e. GON4L). They also assess CRAMP1 binding genome-wide and, together with other genome-wide assays, suggest that CRAMP1 is recruited to

accessible histone promoters when these are non-methylated.

As noticed by the journal and the authors of this manuscript, Matthews et al., and Ingham et al., Mol Cell, 2025 already described the role of CRAMP1 as a regulator of histone H1 in their publications. Moreover, Matthews et al., also highlights the relationship of CRAMP1 with linker histone H1 and PRC2 through the analysis of DepMap data (Fig 1E of their manuscript). Therefore, the major finding of this manuscript has already been described. However, considering that Bodner et al. is scooping-protected following the rules from EMBO Reports, my assessment will not take these publications into account.

The key finding of Bodner et al. is the discovery of CRAMP1 as a regulator of H1 and its association with the HLB and HLB-related factors. The manuscript is of general interest to the molecular biology community, as it addresses general principles of nuclear function. Overall, their claims are supported by their experiments but there is missing information in the Methods section. It would also be good to include the files resulting from their analysis (further specified below). I would recommend its publication after revision of the comments below.

Major comments

1. The authors have multiple datasets (e.g. CRISPR screen results, AP-MS). Could the authors provide the actual data so that it can be a resource for people to look for specific proteins?

All the data for the RNA seq and ChIP-seq will be submitted to the GEO data base so that is freely accessible. As part of the revised manuscript, we have included the full dataset from the CRAMP1-V5 IP-MS experiments as table EV1.

2. In a few instances (e.g. PRC2 components and NPAT), the authors describe specific proteins and their significance in their assays but the dots corresponding to them are not labelled in the Figures. Could the authors please label which dots correspond to them in the CRISPR screen and the AP-MS plots (Fig. 1 and 2) so that one can assess its significance/effect?

In the revised manuscript we have labeled the PRC2 complex in the Fig. 1C and NPAT in figure 1D.

3. Regarding the analysis of significance in the CRISPR screens and in the AP-MS,

information is missing on the analysis. For the CRISPR screens, I cannot find a detailed paragraph in the methods section stating what they did exactly and which threshold/cutoff criteria they used for the analysis. For both analyses, while the plot shows color-coded dots, I cannot find the actual threshold values for the log₂FC and corrected p-values that they use. Could they please provide this information in the text/methods/figures?

Threshold p-values and Log₂FC values have been added to IP-MS experiments and RNA-seq experiments and these values are stated in the figure legends.

4. The fact of not finding some proteins as significant across the different datasets could hint towards having a too stringent criteria for significance. E.g. 'NPAT was also found in our AP-MS dataset, it did not meet our cutoff criteria for significant enrichment'. It would be good to know how far from their cutoff criteria NPAT was, as it could be a matter of looking into what threshold would be most suitable for the data in hand.

NPAT shows a log₂ fold change of 2.1, but failed our p-value cutoff, with a p-value of 0.12. It may be that NPAT is not directly associated with CRAMP1 and its association is controlled by other variables, leading to increased variability compared with GON4L and H1. How NPAT is associated with CRAMP1 in the cell will require significant additional experiments that are beyond the scope of the current study. For the current study, we feel that the chosen cutoffs were useful to focusing the primary interacting proteins of CRAMP1.

5. The authors alternate between using RPE1hTERT cells and HEK293 cells. In some cases, there is some reasoning behind why they use one or the other but it is not always clarified. Can the authors state clearly why they use one or the other in each case? It could be confusing for the reader the jump between the two.

We have clearly state in the text which cell types are used for each experiment. In addition, we have demonstrated that the CRAMP1 loss leads to the reduction of H1 expression in both HEK293T and RPE1-hTERT cells. So we fell that using these cell types for different experiments is a valid approach. In addition, we have provided a rationalization for cell type selection

6. 'CRAMP-1 degradation led to a reduction in H1 protein as quickly as 1day after the addition of dTAG13' - Figure 2F only shows 1 day as the minimum time explored. Saying 'as quickly as' implies that they would have also checked before 1 day for this

specific assay. It could be already degraded before. From the RNA tracks from TT-seq, you can see downregulation already at 6h. Have the authors also checked for CRAMP1 degradation at 6h? This part could also be solved by rephrasing their statement.

Thank you for this comment. In the revised manuscript we have edited the text and included an immunoblot that shows that CRAMP1 degradation occur very shortly after dTAG addition (Fig. 2G).

7. Regarding the results concerning Fig. 2F. Have the authors assessed or have comments on potential toxicity/off-target effects following degradation?

The advantage of the dTAG system, over siRNA or shRNA approaches, is that a large number of publications have demonstrated that the approach is highly specific.

8. The authors explore the relationship of GON4L and CRAMP1 by checking their interaction through specific domains and by testing whether GON4L binding to promoters is affected by CRAMP1 depletion. With this, they suggest that 'GON4L lies upstream of CRAMP1 recruitment to the HLB and histone H1 promoters' ◊ While histone H1 promoters locate to the HLB, the HLB is a structure observed with microscopy. It would be important to provide an image of GON4L staining after depletion of CRAMP1 to support their conclusion.

We feel that the ChIP-PCR experiments are the best approach to determine the localization of CRAMP1 and GON4L to the H1 promoters. And since the GON4L is recruited the promoter of H1 in the absence of CRAMP1, we would not expect any change in the recruitment of GON4L with mutant CRAMP1, so additional IF would not be informative. Although, it is certainly possible that the localization of the protein to the Histone Locus Body may be partially independent of the recruitment to the promoter. However, this would require additional experiments to explain and would be asking a different question than what is posed here.

9. When assessing ATAC-seq, they make the following claim: 'The ATAC-seq analysis shows a correlation between open-chromatin states at the histone H1 genes and the accumulation of CRAMP1 (Fig. 5H,I), suggesting that CRAMP1 is recruited to accessible histone gene promoters.' - at this point, their observations are a correlation and not a functional relationship. An alternative hypothesis is that CRAMP1 could help opening up the chromatin at those points. One way to explore this would be to check ATAC-seq at H1 promoters upon CRAMP1 depletion. However, with the evidence provided, we cannot infer directionality.

We agree that this is a great question that we are excited to explore. We have been careful in the text to not imply a causal relationship between the methylation status of the promoters and the recruitment of CRAMP1 without providing additional data.

10. The authors have these claims:

a. 'We observe that DNA methylation negatively correlates with open chromatin and the presence of CRAMP1 at histone gene promoters in general (Fig. 5J). These findings suggest that DNA methylation may be a factor that regulates cell-type specific histone gene expression, and that only non-methylated H1 promoters are accessible to CRAMP1' ◊ DNA methylation is known as a repressive mark and is anti-correlated with activity. Their claim is speculative at this stage. They would need functional testing or they would need support from previous literature.

b. 'we propose that DNA methylation is the primary mechanism by which histone genes are selected for expression within a cell lineage' ◊ this is speculative. It would need further support.

We agree that it is not currently possible to determine if CRAMP1 binding or methylation status is the primary determinant. We have added that caveat to the discussion section.

Minor comments

11. 'In addition to the H1 genes, CRAMP1 also showed a high degree of correlation with PRC1 components EED, SUZ12, and EZH2.' ◊ These are PRC2 components. Could the authors correct it and also highlight their location in the Figure, as mentioned above?

We thank the review for identifying the error, we have edited the manuscript to identify these proteins as the PRC2 complex.

12. There is a sentence that starts with 'Since CRAMP1 localizes to the HLB' ◊ The evidence for it comes later in the paper, so it does not read right. Could you please rephrase?

This has been rephrased in the revised manuscript... "Since CRAMP1 is associated with two genes known to influence histone gene expression, NPAT and GON4L...."

13. Fig 1G: Can you please add an arrow at the height of the size of the protein?

Arrows have been added to the immunoblot in Figure 1G.

14. 'Full-length CRAMP1 and the deletion mutant lacking the SANT domain efficiently coprecipitated with GON4L (Fig.2D). However, deletion mutants of CRAMP1 that eliminated the D1 or D2 domains of CRAMP1 abrogated the interaction with GON4L. Likewise, full-length GON4L and the deletion mutant lacking the YY1-binding domain efficiently co-precipitated CRAMP1 (Fig.2E).' ♦ should be 3D and 3E

The figure references have been corrected in the revised manuscript.

15. 'Alternatively, deletion of either the D1 or D2 CRAMP1 domain resulted in a protein that was unable to localize to the HLB.' ♦ I would say concentrate instead of 'localize'. It may localize a bit but not enriched.

The text has been edited to reflect the reviewer's suggestion.

16. ChIP-seq was conducted using the RPE1hTERT CRAMP1-V5 expressing cells that show discrete localization of CRAMP1 to the HLB (Fig. 3A). ♦ This is probably figure 4.

The figure references have been corrected in the revised manuscript.

17. Some references are missing - it says REF

The references have been updated. We apologize for the oversight.

18. 'Our identification of the and as the domains responsible for GON4L binding recapitulates recent findings by Ingham et al.⁵² and goes on to demonstrate that the PAH repeats of GON4L provide the interface for CRAMP1 binding' -> this sentence needs rephrasing ('the and').

The text has been edited in the revised manuscript.

Suggestion

It is nice to see the interaction of GON4L and CRAMP1 at multiple levels: physical interaction, protein domains (AlphaFold + depletions + co-IPs), HLB colocalization. However, I find two aspects missing that would round up their characterization and support their conclusions:

1. They do not show the effect of GON4L enrichment at the HLB by imaging upon the

depletion of specific domains (4C). It would be a nice addition to have GON4L staining in 4C, as it would round up their characterization and support their conclusions.

We show by CHIP PCR that CRAMP1 loss does not alter the recruitment of GON4L. Therefore, we expect no change of GON4L concentration at the HLB in response to CRAMP1 loss. We felt that the essential goal was to verify that the HLB was still present, therefore we chose to stain for NPAT.

2. If they have the data, they could show GON4L CHIP-seq in relation to CRAMP1 and NPAT.

We thank the reviewer for the suggestions and agree that knowing the genomic recruitment of GON4L will further our understanding of the mechanism of H1 gene expression. We are working to obtain these data, but were not able to acquire them within the revision timeframe for the current manuscript.

Dear Prof. Foltz,

Thank you for the submission of your revised manuscript. We have now received the enclosed reports from the referees that were asked to assess it, as well as cross-comments. Both referees still have concerns and suggestions that I would like you to address and incorporate before we can proceed with the official acceptance of your manuscript.

I was hoping to receive a more complete revision from you at this point, so that we could proceed with acceptance. Given this is not the case, we can only publish your work next year, which is a pity. Can you please address all remaining concerns carefully now and resubmit a final ms as soon as possible, thank you.

A few editorial requests will also need to be addressed:

- Your ms has 5 main figures but separate results and discussion sections. Please combine both sections to publish your study as a short report. If you prefer to keep the sections separate, please add one more main figure to the ms.

- Please add up to 5 keywords to the ms file.

- The Data Availability Section needs to be moved to before the Acknowledgements.

- Please add a "Disclosure and Competing Interest Statement" to the ms file.

- The REFERENCE format needs to be corrected: it needs to be alphabetical, not numerical; et al needs to be used after 10 author names; DOIs should only be used for preprints and datasets that have not been published yet. Please correct.

- Please co-submit with your final ms a fully completed author checklist, which you can download from our author guidelines <<https://www.embopress.org/page/journal/14693178/authorguide>>. The completed author checklist will also be part of your transparent peer-review file.

- The FUNDING information is not congruent; there is a discrepancy between grant number CA260699 in the ms file and CA26069 in our online submission system; missing in the system: HL153122; the information provided in the Comments box needs to be removed and provided as separate funder via "More Funders" option (a separate entry that can have multiple grants or each grant should be entered as a separate funder). Please correct.

- A callout for Fig 5H is missing, please add.

- Table EV1 is a dataset and should be updated to Dataset EV1 in all places and upldd as a Dataset ms file type. The Appendix S1 file should be updated to Table EV1 (simple table that can be converted to PDF) in all places (including the Reagents and Tools table) and upldd as an Expanded View Content file type. The Appendix can then be deleted.

- In the source data (SD) checklist, the following panels are missing: 1DE, 2H. Also, 2DEFGIJ needs an URL and accession ID in the Data Availability Section given that it is deposited in a public database.

- The manuscript sections should be in the following order: Title page - Abstract & Keywords - Introduction - Results - Discussion - Methods - Data Availability - Acknowledgments - Disclosure Statement & Competing Interests - References - Figure Legends - (Main Tables with legends if applicable) - Expanded View Figure Legends.

- The Methods section should have "Methods" title

- Bounced Email: Justin Bodner - justin.bodner@northwestern.edu; please either remove the author from the system and add him back using the correct email OR send us the correct email so that we can update the account

* Figure Legends - Comments *

- Please define the annotated p values ****/***/**/* as well as provide the exact p-values for the same in the legend of figure 2J, 5G, EV2 D as appropriate and reasonable.

- Please indicate the statistical test used for data analysis in the legends of figures 1D, E; 2D, G; EV2 D

- Please note that the box plots need to be defined in terms of minima, maxima, centre, bounds of box and whiskers, and percentile in the legends of figures 2J, EV2 B, D

- Please note that information related to n (including technical versus biological) is missing in the legends of figures 2D, J; EV2 B-E

- Please note that the error bars are not defined in the legend of figure EV2 C.

- Please note that the measure of center for the error bars needs to be defined in the legends of figures 2B, 5G
- Please note that the red arrow is not defined in the legend of figure EV1 B. This needs to be rectified.

- Please re-write the title and abstract according to the referee comments. Also, the abstract needs to be written in present tense as per journal policy, and it would be good to mention the species the information in the abstract applies to early on.

EMBO press papers are accompanied online by A) a short (1-2 sentences) summary of the findings and their significance, B) 2-3 bullet points highlighting key results and C) a synopsis image that is exactly 550 pixels wide and 200-600 pixels high (the height is variable). The synopsis image should provide a sketch of the major findings, like a graphical abstract. Please note that text needs to be readable at the final size. Please send us this information along with the final manuscript.

Referee #1:

This revised paper has added substantial data to their original paper. They clearly show that CRAMP1 is rapidly depleted within three hours but does not interfere with cell growth, although it does affect the expression of some of the histone H1 genes. There is a small amount of CRAMP1 left in the cell (always will be in this type of experiment) but they provide other previously published evidence by others that CRAMP1 is not an essential gene in cultured cells. There is activation of many PRC-repressed genes, as expected for the known role of histone H1 in repression of heterochromatin. Their data strongly suggests that CRAMP1 stimulates transcription of the histone H1 genes, but is not required for transcription of the H1 genes. Depletion affects expression of both the replication-dependent genes (CRAMP1 is concentrated in the HLB), but also affects expression of the H1-0 and H1-10 genes, the two major replication-independent H1 genes. The title might be changed to reflect its function in the HLB as well as on the H1 genes that aren't in the HLB.

Comments:

1. Introduction. Although it has been called SPT21NPAT by some I would not consider yeast SPT21 to be like NPAT. NPAT(Mxc)'s major role is to organize the HLB, which is present throughout the cell cycle, and it is not associated with any other genes. SPT21 is only required for a subset of histone genes in yeast (Hess... Sternglanz and Winston MCB, Jan. 2004, p. 135-143), and these mRNAs are polyadenylated, and there is no HLB. SPT21 shares the property of being activated by cyclin/cdk as NPAT is activated by Cyclin E/Cdk2, but this activation is to coordinate histone expression with DNA replication, which is activated by the same signals.
2. It would be very helpful to indicate early in the results which H1 genes are expressed in the cells they are using. Is it just H1-0, H1-1 to H1-5 and H1-10? Generally all five of the replication-dependent H1 histones are expressed in cycling cells.
3. Fig. 1F shows that CRAMP1 can pull down NPAT when both are overexpressed. This experiment does not rule out the possibility that the CRAMP1 is interacting with GON4L which is known to form a complex with NPAT. In lower panel of Fig. 5, does the H1-0 antibody interact with all the histone H1 proteins, or is it selective for H1-0? The proteomics data in Fig. 1B shows that CRAMP1 interacts most strongly with the H1-0 and H1-10, both of which are variant H1 proteins.
4. Bottom of page 5, I would say that CRAMP1 is degraded within 3-5 hrs and refer to Fig. 2A and 2H.
5. In fig. 2F what was the time of the treatment with dTAG?. I presume this is from RNA-SEQ data? From the growth curve in Fig. 2B, one would expect all the histones to be down-regulated by 5 days (as the cells definitely must be growing more slowly by then). In the H2B panel of Fig. 2F, there are some H2A genes indicated (likely a typo).
6. Fig. 2H shows the rapid degradation of CRAMP1 convincingly. Including a browser shot of a couple of core histone genes in Fig. 2I would be helpful here, to show the contrast with the H1-2 gene. Also they should say indicate whether the peak heights represent a decrease by indicating whether they normalized the data to the peak heights on other genes that don't change.
7. An important result in this paper is that degradation of CRAMP1 results in a reduction in histone H1 expression but does not abolish it, and the reduction of mRNA is modest (<3-fold). In addition in Fig. 5, they show CRAMP1 is present at the H1-2 promoter and the H1-10 promoter, but not at other H1 promoters including the highly expressed H1-0 gene. The H1-0 gene responds well to CRAMP1 depletion. There is a cell-type difference in this experiment which was done in RPE cells while most other experiments were done in 293 cells, and they don't discuss with H1-0 is expressed in RPE cells. Together these results suggest that CRAMP1 may not be essential for transcription of the H1 genes (although it definitely stimulates it).
8. The DNA methylation data is not developed well and should be deleted from the paper.

Minor Comments

1. paragraph 2 introduction. While some replication-dependent histone genes (e.g. H1-2)3 do produce polyadenylated histone mRNAs under some conditions, the histone mRNAs ending at the stem-loop are "replication-dependent mRNAs" but the polyadenylated forms are not.

Should say "the 3' UTR ends in a well conserved stem-loop structure that binds SLBP. The 3' end is formed by endonucleolytic cleavage by CPSF734, as part of the active U7 snRNP complex, which recognizes a sequence 3' of the SL." NPAT is regulated by CyclinE/Cdk2 phosphorylation.

2. pg 3, end of paragraph 1. I think they mean " a similar repressive role on the core histone genes"; next paragraph should be variants not "variates".

References

1. Kaya-Okur, H.S., Wu, S.J., Codomo, C.A., Pledger, E.S., Bryson, T.D., Henikoff, J.G., Ahmad, K., and Henikoff, S. (2019). CUT&Tag for efficient epigenomic profiling of small samples and single cells. *Nat Commun* 10, 1930. 10.1038/s41467-019-09982-5.
2. Wang, Z.-F., Sirotkin, A.M., Buchold, G.M., Skoultchi, A.I., and Marzluff, W.F. (1997). The mouse histone H1 genes: Gene organization and differential regulation. *J. Mol. Biol* 271, 124-138.
3. Cheng, G., Nandi, A., Clerk, S., and Skoultchi, A.I. (1989). Different 3'-end processing produces two independently regulated mRNAs from a single H1 histone gene. *Proc. Natl. Acad. Sci. USA* 86, 7002-7006.
4. Sun, Y., Zhang, Y., Aik, W.S., Yang, X.C., Marzluff, W.F., Walz, T., Dominski, Z., and Tong, L. (2020). Structure of an active human histone pre-mRNA 3' processing machinery. *Science* 367, 700-703.

Referee #3:

Authors have answered the reviewers comments thoroughly, mostly by improving readability, figure citations, and figure legends accuracy, but no additional data has been included nor experiments/analysis been performed (due to the short time for revision they said).

Fig. 1F legend still talks about NPAT which has been deleted from the figure, and the Results still mentions NPAT was IPed according to Fig. 1F, should be deleted.

Fig. 1G. It is not convincing to me and other reviewers it seems, that H1.0 is being IPed with V5-CRAMP1 from nucleoplasm fraction. In inputs at both figure 1 F and G, several H1 bands are seen, and H1.0 should be the lowest one according to extensive literature, never the higher one that is claimed to be IPed H1.0 in F and is also seen in G chromatin. In nucleoplasm, no H1 band is seen in input, and in IP the marked band also appears in parental control. Further testing with several H1 variant specific antibodies should be made to maintain this part in the manuscript. I don't think figures 1F and 1G are acceptable as it stands.

In methods, TTseq and ATACseq are not explained, this should be added.

No genomic data is publically available so far. GEO numbers should be added to the manuscript before acceptance, and data access be available for reviewers at the time of revision according to my experience as author.

If these issues are solved, the manuscript could be accepted for publication in my opinion.

Cross-comments by referee 1:

The revised paper is more complete than the initial submission.

There is still some sloppy writing and still an incredible number of typos.

There was substantial new data added to the manuscript. Most if not all of the data added was data they already had. There was very little if any new data included that was generated after the first review. For example in Fig. 1 they show the rapid degradation of CLAMP (which occurs within 3-5 hrs after dTAG addition as the now show in Fig. 5), did not affect growth of the cells.

A major thing they should stress is that CLAMP1 stimulates but is not essential for histone H1 transcription, and the cells survive and grow normally (which to me was very surprising). They did not mention this previously.

I agree with reviewer 3 that the binding of CRAMP1 to H1 proteins is not convincing in the protein gels shown. H1 clearly scored as binding to CLAMP in the proteomics. They say the antibody was supposed to be to H1.0, but I guess it may cross-react with other H1 proteins.

In Fig. 2I they show a reduction in nascent transcription of H1.2 and H1-0. They need to show a couple control screen shots of the core histone genes, and explain briefly how they normalized these.

To comeback to Fig 1F and 1G, Fig. 1F is not at all convincing, and could be deleted; it seems to be total nuclear protein sample, sol.

Fig. 1G is more convincing that CARAMP1 may bind histone H10 which is present in the nucleoplasmic fraction. Assuming the upper band are the other H1 histones which cross-react with the H10 antibody this experiment provides evondece that H10 binds to CRAMP1 which is not in chromatin. The bioloigcal significance it not clear.

Cross-comments from referee 2:

I've read referee#1 comments and I think they can be sent to authors, together with mine. I sustain that no new data has been added, only a growth curve as mentioned by referee 3. Basically, figures in resubmission are the same as in first submission, with some rearrangements. All through authors responses to reviewers 2 to 4, they say that no new data has been added due to time constraints.

I confirm my review and the need to make data accessible through GEO.

We thank the reviewers for the constructive feedback and sincerely appreciate their efforts that have improved the manuscript considerably. Please find below a point-by-point description of changes made to the manuscript in response to your comments.

Referee #1:

This revised paper has added substantial data to their original paper. They clearly show that CRAMP1 is rapidly depleted within three hours but does not interfere with cell growth, although it does affect the expression of some of the histone H1 genes. There is a small amount of CRAMP1 left in the cell (always will be in this type of experiment) but they provide other previously published evidence by others that CRAMP1 is not an essential gene in cultured cells. There is activation of many PRC-repressed genes, as expected for the known role of histone H1 in repression of heterochromatin. Their data strongly suggests that CRAMP1 stimulates transcription of the histone H1 genes, but is not required for transcription of the H1 genes. Depletion affects expression of both the replication-dependent genes (CRAMP1 is concentrated in the HLB), but also affects expression of the H1-0 and H1-10 genes, the two major replication-independent H1 genes. The title might be changed to reflect its function in the HLB as well as on the H1 genes that aren't in the HLB.

The title has been revised to "*Distinct Transcriptional Control of Histone H1 Expression by human CRAMP1*"

Comments:

1. Introduction. Although it has been called SPT21/NPAT by some I would not consider yeast SPT21 to be like NPAT. NPAT(Mxc)'s major role is to organize the HLB, which is present throughout the cell cycle, and it is not associated with any other genes¹. SPT21 is only required for a subset of histone genes in yeast (Hess... Sternglanz and Winston MCB, Jan. 2004, p. 135-143), and these mRNAs are polyadenylated, and there is no HLB. SPT21 shares the property of being activated by cyclin/cdk as NPAT is activated by Cyclin E/Cdk2, but this activation is to coordinate histone expression with DNA replication, which is activated by the same signals.

We understand that although SPT21 has some properties that relate it to NPAT, they are not functionally identical. Since the paragraph was discussing primarily metazoan histone gene expression, we have chosen to forego discussion of the yeast SPT21 gene function and this has been removed from the revised manuscript.

2. It would be very helpful to indicate early in the results which H1 genes are expressed in the cells they are using. Is it just H1-0, H1-1 to H1-5 and H1-10? Generally all five of the replication-dependent H1 histones are expressed in cycling cells.

For HEK293T cells, we explicitly demonstrate and state in the text which H1 variants are expressed in Figure 2E (H1-0,-2,-3,4 and -10). Unfortunately, we do not have that analogous information for RPE1 cells. Therefore, we cannot comment on this with respect to figure 1.

3. Fig. 1F shows that CRAMPED can pull down NPAT when both are overexpressed. This experiment does not rule out the possibility that the CRAMPED is interacting with GON4L which is known to form a complex with NPAT. In lower panel of Fig. 5, does the H1-0 antibody interact with all the histone H1 proteins, or is it selective for H1-0? The proteomics data in Fig. 1B shows that CRAMPED interacts most strongly with the H1-0 and H1-10, both of which are variant H1 proteins.

We agree that the NPAT interaction seen in the IP-MS could represent the association of CRAMP1 with NPAT via GON4L. We state in the text this may be indirect and occur through GON4L binding.

We have removed panels 1F and G per the reviewer #2 suggestion.

4. Bottom of page 5, I would say that CRAMP1 is degraded within 3-5 hrs and refer to Fig. 2A and 2H.

We have made the suggested edit to the text of the manuscript.

5. In fig. 2F what was the time of the treatment with dTAG?. I presume this is from RNA-SEQ data? From the growth curve in Fig. 2B, one would expect all the histones to be down-regulated by 5 days (as the cells definitely must be growing more slowly by then). In the H2B panel of Fig. 2F, there are some H2A genes indicated (likely a typo).

The time of dTAG treatment (24 hrs) is noted in the revised text.

We observe no significant changes in cell growth upon CRAMP1 degradation, in HEK293T cells.

The gene names have been corrected in figure 2F.

6. Fig. 2H shows the rapid degradation of CRAMP1 convincingly. Including a browser shot of a couple of core histone genes in Fig. 2I would be helpful here, to show the contrast with the H1-2 gene. Also they should say indicate whether the peak heights

represent a decrease by indicating whether they normalized the data to the peak heights on other genes that don't change.

We have edited Figure 2I to include track examples for two core histones. This augments the complete dataset in figure EV2F which includes all the expressed histones. In addition, we have indicated the peak range for each track. Tracks show CPM normalized read counts derived from the nf-core/rna-seq pipeline.

7. An important result in this paper is that degradation of CRAMP1 results in a reduction in histone H1 expression but does not abolish it, and the reduction of mRNA is modest (<3-fold). In addition in Fig. 5, they show CRAMP1 is present at the H1-2 promoter and the H1-10 promoter, but not at other H1 promoters including the highly expressed H1-0 gene. The H1-0 gene responds well to CRAMP1 depletion. There is a cell-type difference in this experiment which was done in RPE cells while most other experiments were done in 293 cells, and they don't discuss with H1-0 is expressed in RPE cells. Together these results suggest that CRAMP1 may not be essential for transcription of the H1 genes (although it definitely stimulates it).

We have edited the manuscript to highlight the regulatory nature of CRAMP1 and its role in H1 expression, rather than an essential role in histone H1 expression. We have commented in the RNA-seq section that CRAMP1 degradation does not eliminate H1 RNA suggesting it plays a stimulatory role in H1 expression.

8. The DNA methylation data is not developed well and should be deleted from the paper.

We have removed the DNA methylation data as it relates to the core histones (Figure EV5). However, given the established role of DNA methylation in regulating gene expression, and the strong correlation that we observe between the open chromatin state with CRAMP1 binding and low DNA methylation, we feel this helps define the characteristics of active H1 loci, to which CRAMP1 is binding. Please see the response "Referee #1 Email addendum" below for further details.

Minor Comments

1. paragraph 2 introduction. While some replication-dependent histone genes (e.g. H1-2)³ do produce polyadenylated histone mRNAs under some conditions, the histone mRNAs ending at the stem-loop are "replication-dependent mRNAs" but the polyadenylated forms are not.

Should say "the 3' UTR ends in a well conserved stem-loop structure that binds SLBP. The 3' end is formed by endonucleolytic cleavage by CPSF734, as part of the active U7

snRNP complex, which recognizes a sequence 3' of the SL." NPAT is regulated by CyclinE/Cdk2 phosphorylation.

We have edited the introduction to reflect the reviewer's suggestions.

2. pg 3, end of paragraph 1. I think they mean " a similar repressive role on the core histone genes"; next paragraph should be variants not "variates".

Thank you, the edits have been added to the revised manuscript.

References

1. Kaya-Okur, H.S., Wu, S.J., Codomo, C.A., Pledger, E.S., Bryson, T.D., Henikoff, J.G., Ahmad, K., and Henikoff, S. (2019). CUT&Tag for efficient epigenomic profiling of small samples and single cells. *Nat Commun* 10, 1930. 10.1038/s41467-019-09982-5.

This reference was not added since we are not using Cut&Tag

2. Wang, Z.-F., Sirotkin, A.M., Buchold, G.M., Skoultchi, A.I., and Marzluff, W.F. (1997). The mouse histone H1 genes: Gene organization and differential regulation. *J. Mol. Biol* 271, 124-138.

3. Cheng, G., Nandi, A., Clerk, S., and Skoultchi, A.I. (1989). Different 3'-end processing produces two independently regulated mRNAs from a single H1 histone gene. *Proc. Natl. Acad. Sci. USA* 86, 7002-7006.

4. Sun, Y., Zhang, Y., Aik, W.S., Yang, X.C., Marzluff, W.F., Walz, T., Dominski, Z., and Tong, L. (2020). Structure of an active human histone pre-mRNA 3' processing machinery. *Science* 367, 700-703.

Thank you, References have been added.

Referee #3:

Authors have answered the reviewers comments thoroughly, mostly by improving readability, figure citations, and figure legends accuracy, but no additional data has been included nor experiments/analysis been performed (due to the short time for revision they said).

Fig. 1F legend still talks about NPAT which has been deleted from the figure, and the Results still mentions NPAT was IPed according to Fig. 1F, should be deleted.

The reference to figure 1F has been removed.

Fig. 1G. It is not convincing to me and other reviewers it seems, that H1.0 is being IPed with V5-CRAMP1 from nucleoplasm fraction. In inputs at both figure 1 F and G, several H1 bands are seen, and H1.0 should be the lowest one according to extensive literature, never the higher one that is claimed to be IPed H1.0 in F and is also seen in G chromatin. In nucleoplasm, no H1 band is seen in input, and in IP the marked band also appears in parental control. Further testing with several H1 variant specific antibodies should be made to maintain this part in the manuscript. I don't think figures 1F and 1G are acceptable as it stands.

Figures 1F and G have been removed from the current manuscript.

In methods, TTseq and ATACseq are not explained, this should be added.

Methods for both ATAC-seq and TT-seq have been added to the revised manuscript.

No genomic data is publically available so far. GEO numbers should be added to the manuscript before acceptance, and data access be available for reviewers at the time of revision according to my experience as author.

We apologize, however, the US government shutdown precluded us from uploading the data to GEO for the previous revision. The data are now uploaded to GEO.

If these issues are solved, the manuscript could be accepted for publication in my opinion.

Cross-comments by referee 1:

The revised paper is more complete than the initial submission.

There is still some sloppy writing and still an incredible number of typos.

There was substantial new data added to the manuscript. Most if not all of the data added was data they already had. There was very little if any new data included that was generated after the first review. For example IN Fig. 1 they show the rapid degradation of CLAMP (which occurs within 3-5 hrs after dTAG addition as the now show in Fig. 5), did not affect growth of the cells.

A major thing they should stress is that CLAMP1 stimulates but is not essential for histone H1 transcription, and the cells survive and grow normally (which to me was very surprising). They did not mention this previously.

The manuscript does not claim that CRAMP1 is essential. However, we have edited the manuscript to highlight the regulatory nature of CRAMP1 and its role in H1 expression.

I agree with reviewer 3 that the binding of CRAMP1 to H1 proteins is not convincing in the protein gels shown. H1 clearly scored as binding to CLAMP in the proteomics. They say the antibody was supposed to be to H1.0, but I guess it may cross-react with other H1 proteins.

Figures 1F and G have been removed from the current manuscript.

In Fig. 2I they show a reduction in nascent transcription of H1.2 and H1-0. They need to show a couple control screen shots of the core histone genes, and explain briefly how they normalized these.

We have edited Figure 2I to include track examples for two core histones. This augments the complete dataset in figure EV2F which includes all the expressed histones. In addition, we have indicated the peak range for each track. Tracks show CPM normalized read counts derived from the nf-core/rna-seq pipeline.

To comeback to Fig 1F and 1G, Fig. 1F is not at all convincing, and could be deleted; it seems to be total nuclear protein sample, sol.

Figures 1F and G have been removed from the current manuscript.

Fig. 1G is more convincing that CARAMP1 may bind histone H10 which is present in the nucleoplasmic fraction. Assuming the upper band are the other H1 histones which cross-react with the H10 antibody this experiment provides evondece that H10 binds to CRAMP1 which is not in chromatin. The biological significance it not clear.

Figures 1F and G have been removed from the current manuscript.

Cross-comments from referee 2:

I've read referee#1 comments and I think they can be sent to authors, together with mine. I sustain that no new data has been added, only a growth curve as mentioned by referee 3. Basically, figures in resubmission are the same as in first submission, with some rearrangements. All through authors responses to reviewers 2 to 4, they say that no new data has been added due to time constraints.

I confirm my review and the need to make data accessible through GEO.

Referee #1 Email addendum

There are two pieces of data shown for DNA methylation. One shows all the H1 genes. The other shows all the core histone genes (individually I think in EV5). They need to indicate the transcript below the diagram (as a bar); All the transcripts for a particular gene of each class, except the variant genes with introns, are essentially all the same length. and I don't think any H1 genes have introns.

Figure 5 H shows heatmaps that are centered on the transcriptional start sites as we are focused on CRAMP1 binding within this region. The transcripts for individual histones vary in length, so annotating the transcripts for all H1 genes is not feasible. We have removed Figure EV5.

What is the expression level of each gene [they have this data from their RNA Seq data). I think it may correlate but they should say that if it is true (or just give a number of per cent total H1 mRNA on each gene in the diagram..

Unfortunately, we conducted the DNA methylation analysis in RPE1-hTERT cells in order to be able to compare this data to ChIP-seq data, while RNA-seq was conducted in HEK293T cells where the CRAMP-dTAG was constructed. Therefore, we cannot directly compare the RNA levels with the DNA methylation levels. However, we did conduct ATAC-seq in the RPE-hTERT cells, which is a good surrogate for RNA expression. The data in figure 5H (individual H1 genes) & 5I (open versus closed H1 promoters in bulk) demonstrate that open chromatin has significantly lower levels of DNA methylation. We feel this is analogous to showing the RNA levels correlate with DNA methylation as suggested.

How many potential sites (CG doublets) are there in each H1 gene (or 5' flanking region, transcript and 3' flanking region. They could easily show that.

In figure 5 we have added the relative proportion of CpG dinucleotide sequences within the promoter of each histone H1 gene, showing that potential CpG availability does not drive the levels of CpG methylation

They also need to indicate what are the histone genes in Fig.EV5. Are they only replication dependent? or are some variant histones. Replication-dependent histone genes have a relatively high level of C G doublets in their coding region due to their codon usage.

Figure EV5 has been removed.

It is not more experiments. Just providing additional data from experiments they did and looking at experiments others have done on methylation where data for histone genes should be there.

I am not aware of any studies of DNA methylation on histone genes, although the data for histone genes will be in every DNA methylation study and they could download it and look only at the H1 genes to see if there are differences in their cells from other cells that express different sets of H1 genes. If there are differences then that is an interesting observation.

My preference would be to remove the methylation data from this paper since there are no experiments that even hint if it has a function.
That will save them from having to deposit it for others to look at.

We appreciate the reviewer's comments and agree that little is known about DNA methylation at histone loci. We agree that a multi-cell type analysis of existing datasets will be very informative. As we have removed the core histone analysis from this manuscript, we will pursue a comprehensive analysis of DNA methylation at all histone loci in the future.

Prof. Daniel Foltz
Northwestern University Feinberg School of Medicine
Department of Biochemistry and Molecular Genetics
303 E superior St
SQBRC 7400
Chicago, IL 60611
United States

Dear Dan,

I am very pleased to accept your manuscript for publication in the next available issue of EMBO reports. Thank you for your contribution to our journal.

You may qualify for financial assistance for your publication charges - either via a Springer Nature fully open access agreement or an EMBO initiative. Check your eligibility: <https://link.springer.com/journal/44319/how-to-publish-with-us>

>>> Please note that it is EMBO Reports policy for the transcript of the editorial process (containing referee reports and your response letter) to be published as an online supplement to each paper. If you do NOT want this, you will need to inform the Editorial Office via email immediately. More information is available here: <https://link.springer.com/partners/embo-press/editorial-policies#Peer%20review>